# Excavating the Hall of Dreams: The Inventions of "Fine Art" and "Religion" in Japan

**Jason Ānanda Josephson Storm**

Department of Religion and Science & Technologies Studies Program, Williams College, Williamstown, MA 01267, USA; jaj1@williams.edu

**Abstract:** Setting out from Okakura Kakuzō and Ernest Fenollosa's famous "discovery" of the Yumedono Kannon, this article will trace the contested construction of the categories of "religion" (*shūkyō*) and "fine art" (*bijutsu*) in Meiji Japan. In religious studies circles, it has become commonplace to think of "religion" as the only disciplinary master category with issues. However, not only was "religion" invented in Japan, but "fine art" was invented there too. Indeed, categories from "culture" to "society" to "politics" have similar issues. Attending to these will help refocus crucial debates away from an obsession with translation and onto more fundamental issues about "cultural categories" as such. This paper will advance the debate by explaining the attendant constructions of "religion" and "fine art" as process social kinds. In doing so, it will showcase the museum and the temple as central sites of materialized disputation over global categories and their local instantiation. It will show how assimilation to the world-system in the long nineteenth century was a complex multi-generational process of negotiation and contestation, producing new hybrid spaces, returns, transformations, and innovations that then reflected back on global systems, changing them in subtle but profound ways.

**Keywords:** religion; fine art; Japan; art history; Yumedono Kannon; museum; temple; philosophy of social science; translation theory; Okakura Kakuzō; Ernest Fenollosa; Hōryūji

## 1. Introduction

The Japanese art historian Okakura Kakuzō 岡倉覚三 (1863–1913) recounted the much-mythologized discovery of the Bodhisattva of the Hall of Dreams as follows:

> In 1884 [Ernest] Fenollosa (1853–1908), Kanō Tessai 加納鉄哉 (1845–1925) and myself came before the priests of the [Hōryū-ji 法隆寺] temple and asked them to show us the secret buddha (*hibutsu*, 秘仏). The priests said that revealing the secret buddha would definitely cause us to be struck by lightning. In the first year of Meiji, when during the conflict over the mixture of Shinto and Buddhism, the gates to the hall [where the statue was kept] were temporarily opened; suddenly the whole sky was overcast, there was a burst of thunder, and we became very frightened. They decided to proceed no further.

> With this example in mind, the priests did not want to comply with our request. And when we began to open the gates, they were so frightened that they fled. When we opened the hall's gates, almost one thousand years of stench assaulted our nostrils. Brushing aside the cobwebs, we saw a low table of the Higashiyama period. When we cleared this aside, there, directly before us was the sacred statue [known as the Kannon of the Hall of Dreams, Yumedono Kannon 夢殿観音] which measured some eight or nine feet in height. The statue was wrapped in many layers of cloth. Surprised by the presence of human beings, snakes and mice suddenly scampered, frightening us.

> We approached the statue, and when we removed the cloth wrappings there was underneath a covering of white paper. This was the point which had been

reached in the first year of Meiji, when the priests had heard the burst of thunder and suspended the uncovering of the statue.

We saw behind the white paper the serene face of the statue. This was truly one of the greatest pleasures of a lifetime. Fortunately, there was no thunder and the priests were greatly reassured. (Okakura 2001, pp. 57–58. Also reproduced in Takada 1980, pp. 46–47)

There are reasons to be skeptical about the portrayal of superstitious priests and their hidden idols, but in the years that followed, the Yumedono Kannon—and a set of other artifacts housed at that particular temple complex—were the focus of a series of debates about the range and applicability of the categories "religion" (*shūkyō*, 宗教) and "fine art" (*bijutsu*, 美術).

This article will set off from these debates. I want to be especially clear why I find this example relevant and why—in the grand scheme of things—it matters even a little. When Horii Mitsutoshi kindly invited me to contribute to this special issue, I was hesitant because I didn't want to merely rehash arguments I made in my first monograph on *The Invention of Religion in Japan* (Storm 2012), nor did I want to just abbreviate the theoretical insights it has taken me the length of a more recent monograph to explain adequately (Storm 2021). One might argue that is all I've been able to do here (if so *mea culpa*) but after some reflection I thought focusing on the historical constitution of "fine art" might let me show why this work is important not just for scholars of religion but also for art historians.

In the first case, scholars of critical religion scholars have tended to concentrate on the construction of "religion" often in contrast with the secular. This has been valuable work. It can be complexified, however, and even troubled, by looking at how notions of "fine art" and "religion" were fashioned on the ground and in competition with each other. This will show that attempts to focus merely on the peculiarities of the secular-religion binary are insufficient and that most of the conversation about description versus the redescription of "religion" fails to address the most important philosophical issues. To scholars, who think that religion is basically the product of scholarly imposition, the article will also show how the temple and the museum were the major venues for inventing these important categories.

For art historians, I want to suggest the difficult in presuming that the notion of art, and especially fine art, is necessarily a clear-cut concept. In particular, I want to show how importing (and reformulating) a notion of fine art changed things in Japan. Art historians know plenty about the history of the museum and its effect on the national imaginary, but I want to add to their repertoire a set of critical techniques at this conceptual level. In parallel, to scholars of religion who often take the notion of art for granted I also want to give art historians a pause when reflecting on the nature of "religion".

For both groups, I want to explain how objects themselves change meaning depending on the social context in which they are deployed.

## 2. The Invention of "Fine Art" in Japan

On 10 March 1872, an unusual exhibition opened at the Taiseiden Hall 大成殿 of Yushima Seidō 湯島聖堂, a formerly Confucian Temple that had recently been appropriated by the Japanese Ministry of Education. As depicted in a set of three woodblock prints by Ichiyosai Kuniteru 一曜斎国輝 (1830–1874), displayed there were an unusual assortment of curios, including: an archaic globe; a living salamander in a water tank; a statue of a golden *shachihoko* 鯱鉾 fish with the body of a carp and the head of a tiger; paintings of animals and imperial courtiers; examples of calligraphy, textile, jewelry, musical instruments, ceramics, masks, swords, and small sculptures; as well as a child's skull; unusual looking seashells; taxidermized birds, fish, and other faunae.[1]

In many respects, this exhibition was in continuity with Tokugawa precedents. Since 1757, the Yushima Seidō in particular had been hosting displays of man-made and natural curiosities known as *bussankai* 物産会 (literally "product meetings"), *honzōkai* 本草会 ("botanical meetings"), and *hakubutsukai* 博物会 ("public learning meetings"), all of which

were basically just occasions for "gentlemen-scholars" to display their research and related collections of natural and antiquarian curiosities (Aso 2014, pp. 16–17). The *bussankai* was thus part of a broader culture of public display that complemented two other Edo Period exhibition forms: on the one hand, *kaichō* 開帳 "temple fairs" that were typically a temporary exhibition of otherwise sequestered but efficacious relics or images by shines and temples for the benefit of pilgrims and other patrons; and on the other hand, *misemono* 見世物 "carnivals" that often showcased "monsters" and human oddities for public viewing (see Furukawa 1970; Markus 1985).

Nonetheless, the 1872 Yushima Exhibition signaled the beginning of a shift from this older taxonomy in three key ways: First, the Ministry of Education baptized it a "*hakubutsukan*" 博物館. This was a neologism literally meaning "hall of myriad objects", but which had first been used in 1861 by Shogunal delegates to the United States as a translation of the English word "museum"; and indeed, *hakubatsukan* would eventually become the standard Japanese term for "museum" going forward (see Aso 2014, p. 51; McDermott 2006, p. 345; Morishita 2018, p. 5). Second, as hinted by this new vocabulary, many of the objects placed in the Taiseiden Hall were not returned to their original owners at the end of the exposition (as would have been standard for a *bussankai*), but became part of a permanent exhibition. Eventually, this collection would become part of the Tokyo National Museum (*Tokyo kokuritsu hakubutsukan*).

Indeed, even before the 1872 exhibition was over, the Ministry of Education had begun to draw up plans for a system of national museums intended to foster and showcase indigenous arts and crafts (Aso 2014, p. 52). The Meiji Japanese State saw the construction of museums as an important part of the modernization project and crucial to establishing Japan's international reputation (see Tseng 2008). Significant to these policy choices was the background of World's Fairs, in which primarily European and North American nations vied with each other in showcasing their technological, industrial, and artistic innovations. Japan, like other non-Western powers, was originally excluded from submitting to the "art" divisions of these events under the assumption that as an "underdeveloped" nation, it had no real art to contribute. This was a categorization Japanese leaders were necessarily interested in overturning for reasons of national pride and international reputation. Accordingly, the construction of Japanese national museums was intended to be a significant diplomatic step toward being able to participate on equal footing with so-called civilized nations (Aso 2014, p. 31).

The Japanese state, therefore, had a motivated interest in promoting Japanese forms of "fine art". There was a problem, however, because it wasn't immediately clear what might be the indigenous equivalent of this European notion. There were a range of existing terms but they all carried different connotations and implied different distinctions and clustering of objects and techniques. The questions became: What was fine art in Japan? What belonged in fine art museums? What Japanese objects were analogous to European art? Europeans were promoting a distinction between art and craft that relegated many things to the former category precisely because they had been constructed for exhibition rather than use. This distinction did not predominate in Japan and let toward more questions. Did functional objects qualify as art? How about calligraphy? How about pottery? What would word should be used to communicate the category the Japanese state wanted to promote internationally And so on.

To address these compelled the construction of a new vocabulary, that in many respects was tantamount to the invention of fine art in Japan. Indeed, the neologism *bijutsu* 美術 ("fine art"), first appeared in 1872, in an official call for participation in the Vienna International Exposition to be held the following year. *Bijutsu* was a new term combining characters 美 meaning roughly *beauty* and 術 *craft* or *skill* in a rather literal translation of the German *schöne Kunst* and French *beaux-arts* (fine art) that both appeared in the multilingual call. This was a novel vocabulary. To explain the new word *bijutsu* to its readers, the text was glossed with the explanation "music, painting, sculpture, poetry, etcetera are referred to as *bijutsu* in the West" (quoted in Hino 1986, p. 24; see also Aso 2014, p. 46; Kitazawa

2000, esp. 9; Kitazawa 2010). It is worth emphasizing that while paintings, sculptures, and ceramics had existed in Japan for centuries at that point, they had never been understood as part of a central organizing category (Morishita 2018, p. 5). Inventing "fine art" therefore suggested the extensive reclassification of indigenous materials and techniques.

In promoting *bijutsu* as a new taxonomic master category, the Japanese state was also overwriting another category, *geijustu* 芸術 (literally "skillful arts"). As Satō Dōshin 佐藤道信 has observed, this older term was closely associated with Confucian discourse about the six great skills of a gentleman (etiquette, music, archery, horse-riding, calligraphy, and mathematics); and, hence, in the Tokugawa period, *geijutsu* encompassed martial arts, mathematical, musical, and other skills including divination.[2] Hence, Satō speculates that "*bijutsu* was probably coined with the purpose of excluding martial arts and divination, while being limited to those arts relating to a new [imported] notion of beauty" (Satō 1999, p. 5). As Satō's remark suggests, the formation of the category of "art" was in certain respects parasitic on an imported discourse about aesthetics. Indeed, the Early Meiji era saw the translation of Western philosophical works about art and aesthetics (often presented as a new science), which were promoted and debated by elite Japanese cultural translators (for these debates, see Marra 2002).

To put it in slightly exaggerated terms, the formation of *bijutsu* and the accompanying discourse on aesthetics produced a new phenomenological orientation to the objects classified as "art". For instance, Buddhist icons were literally seen in new ways and were judged according to new criteria (Faure 1998, see also Winfield 2013). Purely aesthetic appreciation needed to be learned. It meant that "artwork" was to be apprehended in new ways as embodying transcendental and largely secularized notions of beauty or aesthetic experience.

The invention of fine art had material and financial implications. The new notion of *bijutsu* was popularized in official policies that worked to establish fine-art museums and galleries (*bijutsukan* 美術館), fund art exhibits (*bijutsu tenrankai* 美術展覧会), and even create new public art schools (*bijutsu gakkō* 美術学校). Early in the period, the notion of fine art was often characterized as complementarily to craft by means of the new expression *bijutsu kōgei* (arts and crafts). Nonetheless, increasingly, arts and crafts were seen as no longer on equal footing and *bijutsu* increasingly ascended as the more prestigious category while craft art was relegated to a secondary role. By mid-Meiji, the notion of *bijutsu* had been further narrowed to exclude poetry and literature and even calligraphy, with painting and sculpture—now characterized by way of neologisms *chōkoku* 彫刻 and *kaiga* 絵画—increasingly seen as the core of the fine arts (Graham 2021, p. 2; Lippit 2019, p. 9).

All that is to say, as Aso Noriko has aptly summarized, "The creation of this umbrella category of 'art' also meant the exclusion and demotion of forms of aesthetic production that did not parallel the staples of Western high culture; anything that did not fit the definition of art fell into the category of mere manufactures." (Aso 2014, p. 31). Moreover, while the museum came to supplant older exhibition forms, the Meiji government took particular interest in eliminating the carnivalesque. Thus, along with other attempts by the Japanese government to rein in the "barbaric", regulations were passed against misemono in the early Meiji period: First, in 1870, hoaxes and fraudulent exhibits were prohibited by law; then ordinances in 1872 and 1873 forbade the exhibit of human deformities; and, finally, in 1873, the temporary screen-huts that had been the hallmark of the misemono were banned (Storm 2006, pp. 215–17; Markus 1985, pp. 539–40). Although these measures were detrimental, they did not succeed in eliminating misemono completely. Carnivals continued into the nineteenth and twentieth century, but nonetheless their role had largely been supplanted by national museums (both scientific and artistic).

Although not actively banned, the kaichō (temple fairs) were similarly supplanted insofar as temple collections were reinscribed not as efficacious relics, but as exemplars of Japanese "art". This was the context for the Hōryūji priests' reaction to Okakura and Fenollosa's attempt to gain entrance to the Hall of Dreams. To explain, as Fabio Rambelli has observed, the term "hibutsu" or "secret Buddha" had become popular in esoteric

Buddhist circles in the Kamakura Period (1192–1333), where it was connected to the idea that powerful images should only be made visible, and thus activated, under particular circumstances and in the presence of specific people. To make them visible in other circumstances would be to either cause them to lose their efficacy or invite potential karmic retribution. This was not to say that all hibutsu were always kept unseen, rather that by the Edo period, the display either privately to elite patrons or in the ritual context of a kaichō fair was seen as appropriate to the artifact and central to the ceremonial life of the temple and securing its status vis-à-vis other temples (see Rambelli 2002, esp. 276–77). This is the background that is overlooked in simplistic caricatures of superstitious monks. As Matsuyama Iwao has argued, to allow representatives of the government art commission to apprehend the work would mean that "even if it would be praised as an art object without peer in the world, at that very moment, the Guze Kannon would lose its sacred (*seinaru*, 聖なる) significance." (Matsuyama 1993, pp. 177–78; Rambelli and Reinders 2014, p. 163). Recast, the Hōryūji priests feared that a public exhibition of the hidden relic would at the very least strip it of belief in its spiritual efficacy and in so doing reduce its value to the temple.

Indeed, in many respects, the fate of Hōryūji and its artifacts was exemplary of the impact of the invention of "fine art" (and "religion", see below) in Japan. To sketch some of the relevant history, Hōryūji was said to have been founded in the seventh century and, at the very least, archeological excavations suggest a temple was located at the edge of that site by that period, but it probably only gained significance much later (McCallum 2009, p. 8). The epitaph "Hall of Dreams" first appeared in the 1238 *Shōtoku taishi den shiki* 聖徳太子伝私記 by Kenshin 顕真, which recounted the legend that the Japanese Prince Shōtoku had experienced visions or dreams of a "golden man" who instructed him in the reading of Buddhist sutras. Moreover, it suggested that this legendary golden man had become identified with a specific statue housed in the Hall of Dreams, which by 1106 had already been described as "hibutsu" and sheltered from view. Indeed, the appearance of the statue had become shrouded in mystery, as Kenshin recounts:

> Installed the Hall of Dreams is a gilded image of Guze Kannon, the same size as Prince Shōtoku. Its appearance is unknown now nor was it known in earlier times. Some say it is Prince Shōtoku as a layman carrying an imperial sword; others say it is a two-armed Nyoirin Kannon. (*Shotoku taishi den shiki*, DNBZ 112: 58, translation adapted from Weinstein 1989)

Texts dating from 1362 suggest that the image had been located behind a curtain and that viewing was no longer permitted. Later accounts from 1698 and 1746 simply refer to the statue as a "hibutsu" (Weinstein 1989), while the 1836 *Ikaruga koji benren* 斑鳩古事便覽 by Kakugen erroneously states that "the main icon of the Hall of Dreams is an eleven-headed Kannon, which since ancient times has been a secret buddha with a white cloth shrouding its venerable body (*Ikaruga koji benren* 101, translation adapted from Rambelli 2002). All that is to say, there is no direct evidence that anyone was actually permitted to look at the statue from the 13th century to Okakura and Fenollosa's visit in 1884. However, the Hall of Dreams was if anything considered even more important because of that.

By 1884, Hōryūji had fallen on difficult times. In the preceding era, it had been prosperous due to land grants and a regular stipend provided by the Tokugawa government, but by 1868 its connection to the discredited Shogunate became a liability. The edicts for the separation of buddhas and gods (*shinbutsu bunri* 神仏分離) during 1868–1873 and the attendant confiscation of property and suspension of government salary stripped the temple of most of its lands and sources of revenue. It had been largely able to function independently of parishioners under the old system, but now Hōryūji found itself out of funds and without a popular base of supporters. By 1875, the number of monks housed at the temple had dwindled from over forty living in independent sub-halls to a mere twelve living communally in a single hall in the northeast corner of the compound (McDermott). Moreover, in 1872, a new ordinance by the Ministry of Doctrine forced all Buddhist temples to subordinate themselves to the leadership of one of the seven officially recognized

Buddhist sects. Hōryūji had previously been independent, but it was forced to join the Shingon Sect as a mere branch temple.

In 1876, facing a loss of both income and prestige, the head abbot Chihaya Jōchō (1823–1899) took notice of governmental calls for artistic artifacts and began "donating" items to the national government in return for financial compensation. The Chihaya knew that each object the temple gave away would further reduce its own capacity to attract much needed parishioners and tourists, and made its offerings very carefully (McDermott 2006, pp. 359–60). However, Buddhist artifacts were especially valuable to the Meiji state because they offered an indigenous form "that matched Western notions of fine art in terms of genre (sculpture) and concept (religious) that sidestepped the kinds of Western criticisms aimed at early Japanese efforts to "look" Western" (Aso 2014, p. 31). For that reason, the priests of Hōryūji were unable to resist for long and they had already given much to the national effort by the time Okakura and Fenollosa arrived as inspectors and official experts in the new notion of "fine art". Although today the bodhisattva of the Hall of Dreams remains on the site of the temple, many other objects from Hōryūji have been recontextualized as "art" and permanently relocated to the Tokyo National Museum.

Here is where this is all going; while we could go into more detail about the history of the construction of "fine art" (see especially Satō 1999 and Kitazawa 2000), what I want you to hold onto is the ways in which the invention of "fine art" parallels the invention of religion in Japan.

The Invention of Religion Revisited

By the time this article appears, it will have been about a decade since the publication of my first monograph, *The Invention of Religion in Japan* (Storm 2012). Hence, some reflections on that work and its main argument are overdue. For those who have not read it, the work was an intervention in two related conversations:

First, in many sectors of the academy, "religion" was then and is still now widely taken to be a universal aspect of human experience. Religion is typically supposed to be found in all cultures and through all times, individual atheists and nonbelievers excepted, and it is often assumed that different "religions" serve similar needs and have similar essential properties.

Against this view I argued that the category "religion" is historically conditioned and was consciously formulated often to meet political ends. The main point was not that words have etymologies, but that the emplacement of certain categories transforms their members in demonstrable ways. As I showed, the virtual neologism 宗教 (Jp. *Shūkyō*, Ch. *Zongjiao*, Ko. *Chonggyo*) was coined in Japan in the 1870s as a translation for the Euro-American "religion" and was then popularized throughout East Asia. Defining religion in Japan was a politically charged, boundary- drawing exercise that extensively reclassified the inherited materials of Buddhism, Confucianism, and Shinto. All three traditions were radically changed in a way that has only recently begun attracting scholarly attention. We cannot presume a stable content for "Japanese religions" in any time period, let alone continuity over time. Thus, I aimed to undermine the contention that religion is a natural kind or a cultural universal.

Second, I was attempting to intervene in a separate conversation about the category "religion". Scholars such as Talal Asad (Asad 2003), Tomoko Masuzawa (Masuzawa 2005), J.Z. Smith (Smith 1998), and a host of others had repeatedly argued that "religion" was merely a culturally specific category that took shape among Christian-influenced Euro-American intellectuals and missionaries; and which was then imposed unilaterally on parts of the colonial world. Starting from this orientation, an additional set of academics traced the European construction of various non-European religions, from the *British Discovery of Buddhism* to the Jesuits' *Manufacturing Confucianism.* To be blunt what bothered me about this set of texts was how non-European voices almost completely dropped out (e.g., as though there were no Sri Lankan figures involved in the discovery of Buddhism).

Although allied with this latter critical movement, I wanted to back against the argument that "religion" attained international hegemony without resistance. I wanted

to show how not Europeans, but *Japanese* (albeit elites) invented religion in Japan. To be clear, I emphasized foreign pressures, international networks, and asymmetries of power. Japanese translators and government officials were almost literally staring down the barrels of American and European cannons, but they still managed to exhibit tactical agency or what we might call strategic resistance to these external demands. One of the aims of the book was to demonstrate how one non-Western nation made the discourse of "religion" its own. It argued that in Japan, the concept of religion had to be actively indigenized, and furthermore, this indigenization was undertaken largely by a centralizing government determined to reconfigure the internal constitution of the nation and to shore up its standing in the world. Looking at the formation of religion in Japan not only revealed its global diplomatic contours but also shows how the discourse of religion is woven into the fabric of the nation state and how guarantees of religious toleration function to increase state power and to reconfigure entire cultural systems.

The decade since the publication of *The Invention of Religion in Japan* has a seen a wealth of scholarship in English on the process of constructing the category of "religion" in Japan (including: Hoshino 2012; Isomae 2014; Maxey 2014; Krämer 2015; Horii 2018; Thomas 2019). Although the topic is controversial, scholars in the subfield have increasingly come to recognize that "religion" is not a universal part of human nature, but is a culturally specific category that initially took shape in Western Christendom at the end of the seventeenth century and was then radically transformed through a globalization process connected to European colonialism over the course of the long nineteenth century, producing both "world religions" and discourses around "religion" as an autonomous domain of human experience.

Here is the thing. One could nearly duplicate my argument about "the invention of religion in Japan" and make it about "fine art". "Fine art" emerged in roughly the same period Europeans were coming to construct universalizing notions of "religion". While "religion" typically brought with it a messy taxonomy centered on a Christian propotype. In this case, the new category of "fine art" brought together a collection of classical "arts"—painting, sculpture, music, poetry, dance, and architecture—and this collection of arts was the product of an arbitrary culturally and historically contingent grouping. Discussions of "art" in general were inextricably bound up in an eighteenth-century European shift that repositioned disparate crafts under a common taxonomy as arts, and also produced a new orientation based not on their diverse materials and modes of production but on connoisseurship and "aesthetic" experience (see Storm 2021, pp. 64–65). These were lacking elsewhere. Hence, as I have been arguing here, fine art needed to be invented in Japan, as there was no preexisting term that covered the same set of categories and practices. In summary, "fine art" is in some respects a modern European invention, which had to be actively indigenized elsewhere.

Just as the invention of religion displaced older indigenous taxonomies, the construction of Fine Art in Japan displaced an older category schema around *geijutsu.* The formation of *bijutsu* therefore clustered new things together under a common heading, even as it excluded others. In its displacement of *geijutsu,* it specifically excluded mathematics and divination among other things. The globalization of the category of "fine art" was also rooted in a set of, more or less explicit, binary opposition between "art" and "craft" and between aesthetic and functional objects. As one might suspect, these binaries were asymmetrical, suggesting not merely an analytical bifurcation but a hierarchy of value. All that is to say, the invention of art was based in rhetorical oppositions between art and other adjacent categories, whose suppression or exclusion was essential for the category to assert its own closure.

The invention of "fine art" necessarily involved translation. It resulted in the coining of new terminology, such as *bijutsu* and *bigaku* 美学 ("aesthetics"). It also advanced via new prototypical exemplars, which were typically taken to represent the pinnacle of art. To return to the opening example of this article, according to Fenollosa, the opening of the Dream Pavilion was momentous because the Yumedono Kannon was comparable

to da Vinci's *Mona Lisa*, which he seems to have regarded as an implicit paragon of fine art (see Fenollosa 1921, vol. 1, p. 51; see also Kleitz and Lucore 2019). In this respect, Fenollosa was centering a particular notion of both the formal and functional properties of art and encoding an asymmetrical comparison in his writings about Japanese art history. Scholarship on art was part of the process of producing the category. Indeed, the translation of art saw the construction of a new discourse around aesthetics. This in turn produced a new kind of observer and reified a new type of experience. One can think of the literal importation of the compositional and perspectival rules of painting that had developed in the European Renaissance; or one could imagine the new attitudes of a museum visitor or art collector that comes to see and value works in a particular way. Moreover, "art" in Japan as elsewhere came to function as a normative concept. Succinctly put, to say of a particular work "this is not art" or "this is art" is to render an aesthetic judgement.

But the construction of a notion of fine art in Japan was not merely about translation. It was deeply connected to the diplomatic crucible of the late nineteenth century. As noted above, there were motivated reasons that Japanese elites had to procure indigenous art and to find local analogues that then could be staged in international arenas. The codification of art was connected to new government policy, laws, and institutions. Moreover, the invention of the category of fine art also involved the concrete importation of new materials (e.g., paints and canvases) and new techniques that were then used in subsequent creations.

The invention of fine art literally produced new spaces, museums, and galleries, which functioned as locations for the extensive reclassification and reinterpretation of older materials. Think of how museums cluster objects based on artists, location, or period of composition, and provide new interpretations of them by means of plaques and brochures. This has a transformative effect on the works classified and included (or excluded) from the museum. For instance, think of two different snow shovels from the same manufacturer, one used to shovel snow from the streets and the other purchased by the French Dadaist artist Marcel Duchamp, signed by him, titled *Prelude to a Broken Arm,* and then housed in a museum where it can be "admired" as art.

To put what I have been saying in more Weberian (or perhaps avant-garde) terms, identifying something as "art" functions to separate it out or intensify the conceptual work that the process of differentiation undertakes socially, so that there are now a new set of special practices—legal and governmental, etc.—that pertain to it. Recast, to conceptualize something as "art" is to remove it from the everyday—to separate it from every other kind of social practice, to render it independent and in some sense transcendent. Art is supposed to primarily have an aesthetic rather than a functional role. Ordinary objects are considered tools, not art. Art becomes property of the museum and the gallery, something displayed instead of used (see Storm 2021, p. 65).

In summary, the invention of fine art is an ongoing process. Again (as I argued about "religion"), this is not invention out of whole cloth, but typically the bricolage reconfiguration of pre-existing materials into new relations and new forms. Various entities become (or are reclassified as) art. This is not a teleological or transhistorical process, but one that originated out of a particular logic at a particular moment in Western Europe and America, and its globalization was necessarily selective and to some extent arbitrary. This process is necessarily incomplete and always has a remainder. Moreover, the boundaries of art are always changing.

This unfolding construction of art has concrete implications for the category. Scholars in Religious Studies are likely to know about Wilfred Cantwell Smith's critique of the limits of the category "religion" (Smith 1962). However, they are less likely to know of the philosopher Morris Weitz's earlier critique of the limits of the category of "art". In Weitz's 1956 essay, "The Role of Theory in Aesthetics", he began by noting that aesthetics has always been obsessed with creating various solutions to the question "what is art?" But he contended that this entire method was flawed because it wrongly assumes that works of art share a common essence. When one considers the variety of works commonly regarded as art—e.g., from paintings to ballet, from sculptures to piano, and from concertos

to signed urinals—it is easy to see that there are no necessary and sufficient conditions for membership in the category "art". No matter how you define *art*, the definition will be either too narrow and exclude works commonly recognized as art, or it will be too broad and include things not generally recognized as art. There is no particular set of features or properties that can serve to characterize and distinguish "art" from everything else. The category has no inherent limits. Art cannot be defined. Any attempt to demarcate the limits of art will result in a corresponding artistic movement that aims to transcend or upend those very limits. Artists will always violate the definition of art. Thus, Weitz concluded, a theory of art is logically impossible (Storm 2021, pp. 62–63).

It could almost go without saying but the category of "religious art" is doubly problematic: it presents an inherent tension between an object in a context of worship and the historical appreciation of it. Think of the difference between a statue of a bodhisattva located in a temple and interpreted as an efficacious relic and the same work transposed into a museum and marked and described in formal terms as the product of a particular era and region.

All that is to say, the critique of art bears an uncanny resemblance to that of religion. Again, we see a pattern in which the concept is exposed as modern, culturally relative, normative, and in fundamental contradiction with its area of study. Fine art, in the same manner as religion, had to be invented in Japan. This should also make us uncomfortable as scholars, because for every move in the debate between critical religion and phenomenology of religion, we can find an analogous argument in the philosophy of aesthetics. If the problem was merely with "religion", we could solve it by defaulting to other categories. This would be as though (as in Hasbro's "Taboo") one could do religious studies as long as one merely never mentions the word "religion". This clearly will not work as almost all the other words in our conceptual vocabulary from art to politics to history have similar issues. So it seems that something more fundamental, and more common, is at stake.

The Limits of Translation or Epistemological Anarchy?

Scholars in critical religion have often posed our intervention as primarily an issue of translation. For instance, as Brent Nongbri recounts:

> The historian Edwin Judge suggested a three-step procedure to follow when one encounters the word "religion" in a translation of an ancient text. First, cross out the word whenever it occurs. Next, find a copy of the text in question in its original language and see what word (if any) is being translated as "religion". Third, come up with a different translation: "It almost doesn't matter what. *Anything* besides 'religion'!" (Nongbri 2013, p. 156)

This might seem to suggest that—similarly to the Counter-Reformation assault on vernacular translations of the Bible—the word "religion" is in some sense fundamentally untranslatable. However, as the preceding section about *fine art* implies, religion is not the only category with a similar set of problems. Indeed, scholars outside of religious studies have martialed similar arguments to suggest that terms such as "art", "society", and "culture", and a host of others are also fundamentally untranslatable (for "art" see Dutton 2000, for others see Storm 2021, pp. 71–72). Taken to its logical extension, this *might* seem to suggest that translation is impossible (for a counter argument, see Storm 2021, pp. 183–86). If this were the case, then one might wonder if "religion" and ancillary terms such as "art" actually represent special problems or if scholars have merely managed to be led astray by some more generic features of translation as such.

Approached differently, we seem to have produced a semantic puzzle. To highlight this seeming paradox still further, if you think the scholarly deployment of the category of "religion" is a problem because it lacks a clear analogue in non-European cultures, then one might wonder about the term's diverse meaning within one langue (both temporally and regionally). To explain, according to a classic (albeit probably incorrect) account of linguistic meaning provided by Gottlob Frege, *meaning* is a combination of (1) sense (*Sinn*), that is to say, its description or cognitive significance; and (2) reference (*Bedeutung*), namely, the thing indicated or referred to by the concept (Storm 2021, pp. 58–59). On these grounds,

it would seem that "religion" has had significantly different senses at different periods even within the history of the English language, and very few languages have shared that sense at any given time. Moreover, religion has referred to a shifting set of extensions, and no pre-modern individual and few non-European languages have shared that set of references. Does everybody consider Pastafarianism a religion? Evidently not according to the US state of Nebraska, but it still has some passionate advocates who argue otherwise. So even all US Americans do not seem to share the same sense and reference. However, there might seem to be something suspicious about all this. For instance, Shakespeare's use of "religion" likely presumed a different sense of the word and necessarily suggested a different reference set. So that would mean that the lines "It is religion that doth make vows kept; But thou hast sworn against religion" could not be translated into "religion" in English. To put a point on it, according to this account of meaning and translation, "religion" cannot be translated into "religion". This seems to be a paradox.

To my mind, this implies two related issues. To foreshadow, I will argue that we need a different notion of meaning as well as a different notion of how concepts and social kinds change over time.

In the first case, I think the background theories of meaning and translation in most sectors of the academy are incorrect. At the very least, many of the scholars who have most vociferously argued for the impossibility of translating "religion" and related categories have done so without a clearly articulated account of either meaning or translation. Similarly, the parts of the academy where the impossibility of translation is taken to be a truism have theorized themselves into a strange impasse. Our first clue to this is that they have often argued for translation's impossibility in reference to translated texts (e.g., the English translations of Derrida's *Le Monolinguisme de l'autre* or Walter Benjamin's "*Die Aufgabe des Übersetzers*"). This might suggest that something fishy is going on. Indeed, as I have argued elsewhere, most arguments for the impossibility of translation make their case by way of the linguistic meaning that has been lost in translation, which is itself a translation. To argue that "*Pain et vin*" is untranslatable is often to make the case by providing an account of what meaning has been lost, but this argument is just an additive translation (see de Man 2000). In other words, the impossibility of translation is mostly read in translation and typically staged by way of the very thing that it claims cannot be achieved.

All that is to say, I think that the human sciences need a more productive account of meaning and how translation works, and such an account can be found elsewhere (Storm 2021). However, as important as that is, I argue that it does not itself resolve all the problems with categories such as "religion" and "art". This is in part because, as I have been arguing, a further issue with these categories is not just a problem of translation, but rather that their introduction into a culture that lacks such a concept produces demonstrable and sweeping changes—intellectual, legal, and cultural.

This leads us to the second issue. It transpires that religious studies is not alone in adopting a mode of auto-critique. Gradually since the 1950s, the meanings of the key master categories in a range of disciplines have come to be disintegrated, leaving disciplines in various states of skeptical crisis or flat-out denial. Scholars have observed some combination of the following. First, their key categories are subject to competing definitions (e.g., 164 definitions for "culture") with fuzzy boundaries and gray areas. Second, that they are often rooted in binary oppositions that they cannot sustain (e.g., the putative opposition between "culture" and "nature" seems necessary to sustaining the meaning of both categories but itself cannot be maintained in the face of repeated hybrids, see Latour 1993). Third, they often mask a diversity of lived experiences, local and temporal variations, under a reified abstraction (e.g., what conceivably could "American culture" include that did not either obliterate internal differences or represent a merely bland overgeneralization; where does the boundary of one culture stop and another begin?).

Fourth, the production of the category in question is often the result of vested interests and the mechanisms of power (e.g., *culture* served primarily to position anthropology vis-à-vis other academic disciplines and centrally as part of the project of European colonialism

that presented the other in terms of "primitive culture" as opposed to modern civilization. In other words, essentializing "culture" and "cultural difference" was part of the process of legitimizing European expansion and civilizing missions). Fifth, the category in question has normative force typically including some actors or examples at the expense of others (e.g., "culture" does most of its work by removing something from nature, but then "culture" itself becomes read as a source of essential difference, reinscribing all the things it was meant to displace). Sixth, one can always relativize the category by observing that some non-European culture or cultures, "they don't have our concept of X". What is meant by "our notion" is intimated but not articulated in sloppier forms of this argumentation formula. In more sophisticated accounts, however, after establishing X's semantic field, it is possible to show that no term exists or existed in the non-European language corresponding to the European term or covering anything close to the same range of meanings (e.g., East African languages "have no word for culture". See Wijsen and Tanner 2002, p. 25).

This list is not exhaustive (see Storm 2021). Taken together these suggest a set of techniques that could be remobilized to produce epistemological anarchy in almost any field. However, there is something else significant here. The critiques of all the various disciplinary master categories are similar, and this might seem to suggest that it is because all these categories are similar in some fundamental way. Accordingly, *granting* these critiques enables us to describe—in inversion—central features of how social species (including humans) produce their shared environments. Put differently, they tell us something about how the social world is put together; and the mechanisms through which concepts and social categories are produced and maintained.

In brief, we need to reject the reification of the social world. We need to reject the idea that it consists in clearly delineated categories, describable by means of necessary and sufficient condition definitions, and consisting in roughly fixed cross-cultural and cross-temporal constellations. Succinctly put, we keep deconstructing the categories of our disciplines because we have a mistaken notion of what it means to be a category or kind itself. We have to reject the search for essences and essentialized definitions that has defined the philosophical and often scientific enterprise for more than a hundred years. It is as though we have been looking for objects while we should have been looking for processes. Recast, many of the deconstructive criticisms of the disciplinary master categories amount to identifying errors stemming from reification, atemporality, and misplaced concreteness—in other words, faults rooted in misidentifying the processual nature of their subject matter.

An Alternate Ontology

What does this mean for us as scholars?

A detailed explanation is impossible due to space constraints. I would direct readers elsewhere for further explanation of what I have called, a "process social ontology" approach to "social kinds" (see Storm 2021).[3] But at first pass, many of the deconstructive critiques of the disciplinary master categories boil down to detecting mistakes caused by reification, atemporality, and misplaced concreteness—in other words, flaws caused by misidentifying the processual character of its subject matter. This is not merely a problem with a few terms, as no language is truly exempt from the legacies of power.

Granting all the critiques should therefore encourage us to recognize that the social world is highly varied and constantly changing. It is best thought of as a complex system of interweaving processes and it is these processes that give our categories whatever "identity" they possess. This matters because the prevalent forms of analyzing human affairs are often geared toward substance thinking rather than process thinking. We frequently refer to "colonialism" or "modernity" as if they were bounded objects with distinct borders and even as though they were subjects, rather than dynamic processes. Our major terminology for the disciplinary master categories is still caught up in substance-like language, for example nouns such as *art* and *religion*, and we pose substance-type questions of them, asking for instance "is X a religion?" However, when approaching our subject in this manner, our basic questions often amount to category errors.

Furthermore, although our collective environment is changing continually, it is not all changing at the same rate. Systems of power can generate temporary stabilities and imperfect homogeneities. However, power is not univocal. We can (à la Foucault) "decapitate" the sovereign theory of power, but the result is not merely a uniform or capillary theory. Instead, power functions by means of diverse processes with particular effects. Expressed differently, various aspects of our collective environment are stabilized by diverse types of anchoring processes. These anchoring processes are important because, as I am presuming change and difference, relative stability and similarity become the things to be explained. Hence, we need an account of the multiple distinctive, causal, anchoring, or stabilization *processes* that give various aspects of our world their shared properties.

Elsewhere, I discuss different, if often interwoven, types of anchoring processes, including: A. *dynamic-nominalist*, B. *mimetic*, and C. *ergonic*. This list is not exhaustive. However, in brief: A. *dynamic-nominalist* refers to some aspects of our social world, which are stabilized by classification processes combined with mechanisms of enforcement (cultural, social, institutional, legal, and so on). Examples are bountiful: legal processes that confer particular capacities on certain groups (such as the ability to vote), which have built-in mechanisms to attempt to ensure compliance; boundary policing that goes on in academic disciplines; social norms of shaming or reciprocity that encourage or prevent certain kinds of behaviors; tax codes that provide religious exemptions to organizations that take on properties, and so on (this departs from and complicates, Hacking 1999, 2002). B. *mimetic* means that some aspects of our collective environment are stabilized because of copying processes combined with a consistent environment that tends to stabilize or limit variation (this departs from and complicates, Millikan 2006, 2017). C. *ergonic convergence.* Sometimes kinds share properties through a process of selection or design intended to fulfill a certain function. This is meant in part to cover an insight emerging from Herbert Simon and others (Simon 1996), which states that artifacts are best understood in terms of their purpose or teleology. In other words, if you know that something is a thermostat you can produce more robust generalizations about it than you could merely knowing the material out of which it has been composed. This also why certain kinds of artifacts and social kinds exhibit what amounts to Weberian rationalization (e.g., there is a tendency for "armies" to become better at fighting).

Again, I want to emphasize that these anchoring processes are often entwined in any given case. However, thinking in terms of these kinds and how they have been entwined nonetheless has explanatory value. For instance, when ergonic convergence becomes entangled with dynamic-nominalism, it can exhibit what is known as Goodhart's law, often summarized as "when a measure becomes a target, it ceases to be a good measure". When classification and teleological function become enmeshed, the kind in question tends to exhibit rationalization toward the measure, which then means that the measure itself is no longer valuable for its original purpose.

The other point I want to emphasize is that these aspects of our collective environment are not identical to the terms we have for them. Social kinds exceed their reference, and research in the human sciences often proceeds by discovering hitherto unknown properties. For example, "traffic jams" became a problem before there was a common term for them. It also means I can competently discuss "traffic jams" without knowing exactly what defines a traffic jam, and so on. It might even transpire that "traffic jams" have unanticipated properties that no one knows about (e.g., there might be a way to cause or prevent traffic jams that at present no civil engineer has yet discovered). Moreover, disciplines frequently refer to what amounts to the same kind in differing terminology. Shared terms can also mask significant conceptual divisions (e.g., different people mean different things by the term "religion"). All that is to say, terms factor into social kinds but are not identical to them.

### 3. The Hall of Dreams Revisited or Bodhisattva Reburied?

Having begun with Okakura Kakuzō's account of the Dream Hall, I'd like to return to Ernest Fenollosa's description of the same event:

> This most beautiful statue, a little larger than life, was discovered by me and a Japanese colleague in the summer of 1884. I had credentials from the central government which enabled me to requisition the opening of godowns and shrine. The central space of the octagonal Yumedono was occupied by a great closed shrine . . . On fire with the prospect of such a unique treasure we urged the priests to open it by every argument at our command. They resisted long, alleging that in punishment for the sacrilege an earthquake might well destroy the temple. Finally we prevailed, and I shall never forget our feelings as the long disused key rattled in the rusty lock. Within the shrine appeared a tall mass closely wrapped about in swathing bands of cotton cloth, upon which the dust of ages had gathered . . . at last the final folds of the covering fell away, and this marvelous statue, unique in the world, came forth to human sight for the first time in centuries . . . But it was the aesthetic wonders of this work that attracted us most. From the front the figure is not quite so noble, but seen in profile it seemed to rise to the height of archaic Greek art . . . But the finest feature was the profile view of the head, with . . . lips, on which a quiet mysterious smile played, not unlike Da Vinci's Mona Lisa's. (Fenollosa 1912, pp. 50–51)

Perhaps the most striking thing about this excerpt is Fenollosa's compulsive need to situate the work within a notion of "aesthetics" that he articulates vis-à-vis comparison to Greek and European art. The statue's Buddhist symbolism, the reasons the priest's kept the work cloistered, and its potential efficacious powers, have all been dismissed as superstitious or at best irrelevant. The Bodhisattva of the Dream Hall has been reduced to a formal description of artistic attributes and values such that the most important thing about it is how it is seen, experienced, by Fenollosa himself as fine art connoisseur. He does not depict this interpretation as an imposition but rather as discovery of something already present. Yet, Fenollosa does not even situate these aesthetic features with reference to East Asian artistic models. He seems to be suggesting that the statue is valuable because it resembles the Mona Lisa (even though the statute obvious has no connection to that work). In that respect, it illustrates the invention of "fine art" I have been pursuing in this essay, especially insofar as the supposedly aesthetic universal is transparently a European ideal staged (in this case over against the wishes of the Japanese owners of the work).

The other conspicuous thing about this passage is that Fenollosa's "Japanese colleague" goes unnamed. Fenollosa would not be the first or last American ethnographer to ignore or minimize their native interlocutors. But that said the relationship between Fenollosa and Okakura was significant for both the invention of fine art and, to a lesser extent, the invention of religion in Japan.

In the first case, Fenollosa and Okakura both had a substantial impact on the construction of the notion of fine art in Japan and then the promotion of a notion of Japanese art on the international stage. This was not merely a result of their widely read publications, but as I have been arguing necessarily had institutional dimensions. Both Fenollasa and Okakura were crucial to the establishment of the Tokyo University of Fine Arts 東京藝術大学 as well as the Tokyo Imperial Museum. At different times, both men would take posts as curators of "Oriental Art" at the Museum of Fine Arts in Boston. Fenollosa was also involved in helping draft Japanese legislation for the protection of indigenous artwork and helped chose the art that presented Japan at the World Columbian Exposition of 1893. All that is to say, there were both involved in establishing and policing the boundaries of fine art.

The second case—their role in the construction of religion—is less clear cut, but is still worth discussing nonetheless. As I have argued elsewhere (Storm 2012, 2017) the very moment that oppositional categories like religion and fine art are produced their opposition

cannot fully sustained. There will almost necessarily be crossings and remainders. Thus, in the very act of striping art of its connection to Buddhism and reifying the aesthetic, Fenollosa and Okakura attempted to suture the gap between religion and art.

An extended digression will explain. In *The Ideals of the East* (1903)—the work that made Okakura famous to an international audience—he fairly explicitly charts the history of Japanese art according to a Hegelian model. For instance, Okakura writes: "The conquest of Matter by the Spirit has been always the purpose of the striving of world-forces . . . " (Okakura 1970, p. 163). Here he is blatantly reiterating the central pillar of G.W.F. Hegel's philosophy, namely that all history is the unfolding of *Geist* (spirit/mind) in over (and through) matter. Yet, Hegel had famously insisted on the peak orientalist claim that history moves from East to West. So, even while he was adopting the Hegelian model, Okakura argued in contrast that Spirit had not left Asia behind but rather that the "East" had transitioned through the exact same phases–the Symbolic, the Classical, and the Romantic– in art history that Hegel had outlined in his philosophy of aesthetics (compare Okakura 1970, p. 164 and Hegel 1971). For Hegel, Romanticism represented the final stage in the history of art, because in this phase humanity recognizes the infinite within while simultaneously in the religious sphere the Geist (basically God) becomes too abstract to be represented (see Desmond 1986). Thus, art is dead because it no longer works as a method for the comprehension of the absolute Geist.

This is where Okakura again departs from Hegel. Okakura argues that while in the West Romanticism became increasingly objective and materialistic, in Japan Romanticism (which he says blossomed in the Ashikaga period) became focused upon subjectivity and idealism (Okakura 1970, p. 166). According to Okakura, in the Ashikaga period, Zen Buddhism simultaneously individualized and internalized the understanding of the Spirit (Okakura 1970, pp. 172–73). This enabled Japanese art to continue, in close connection with Buddhism, as a method for apprehending the Spirit. For Okakura, while Western art is dead, Japanese art continues to be a vehicle for the representation of the Spirit and consequently is still a valid method for apprehending the absolute.

Interpreting Buddhism along these lines leads to a strange version of Buddhism, in which Okakura attempts to outdo Hegel in his own terms. Okakura describes Buddhism as an explicit attempt to embody self-understanding and freedom–the two key aspects of Hegel's concept of Geist For example, when Okakura is discussing the Zen Buddhism of the Ashikaga period he writes: "The human soul, to these [Zen] thinkers, was itself the Buddhahood in which the universal, as manifested in the particular became resplendent . . . Thus their training was centered on the methods of that self-control which is the essence of true freedom." (Okakura 1970, p. 171). Crucially, Okakura is transforming Hegel's description of a world force into a Zen practice. This is a claim he reiterates elsewhere, describing "the Buddhist theory of evanescence and its demands for the mastery of spirit over matter . . . " (Okakura 1906, p. 90) and asserting that: "The ideal [Buddhist] monk is the child of freedom, who, dying to the mundane is reborn in the realm of the spirit." (Okakura 1904, p. 57). Again, these quotations exhibit the same Hegelian terminology now being (supposedly) discovered in Buddhism.

The irony of describing a Buddhist monk as a "child of freedom" seems to be lost on Okakura. But later in the same text, he asserts that "True spirituality forsook the luxury of the monastery . . . to take its rugged seat in the breast of the lonely ronin-scholar." (Okakura 1904, p. 69). So in this respect Okakura is attempting to reformulate a lay, and even anti-monastic, version of Buddhist practice. He is also romanticizing a "ronin-scholar" as the ideal practitioner of both Buddhism and freedom. That sounds a lot like a self-portrait. In his private life Okakura practiced forms of esoteric meditation that had previously been reserved for the monastic elite (see Chi and Kakuso 1923, p. 110). He also promoted the arts as religious practices in the broadly Hegelian terms we should now have come to expect. His famous writings on the tea ceremony are merely a further extension of the same impulse (see for example, Okakura 1906, pp. 80–81).

The surprise, perhaps, is that while Fenollasa criticized those who understand Buddhism as "theosophy mixed with a little diluted Hegel" ([Chisolm 1963](), p. 104) like Okakura he actually advocated a form of Buddhism that seemed to do just that. Fenollosa exhibits this in the introduction to his collection of poetry (*East and West*):

> The synthesis of two continental civilizations [East and West], matured apart through fifteen hundred years, will mark this close of our century as an unique dramatic epoch in human affairs. At the end of a great cycle the two halves of the world come together for the final creation of man . . . The violence of the West has been softened by the feminine faith of love, renunciation, obedience, salvation from without . . . One the other hand, the peaceful impotence of the East has been spurred by *her martial faith of spiritual knighthood, self-reliance, salvation from within. The intense individuality of her esoteric discipline upholds the fertile tranquility of her surface.* This stupendous double antithesis seems to me the most significant fact in all history. The future union of the types may thus be symbolized as a twofold marriage . . . Within the coming century *the blended strength of Scientific Analysis and Spiritual Wisdom should wed for eternity . . . Spirit achiev[ing] its sublime purpose of uniting the East and the West.* ([Fenollosa 1893](), pp. v–vii Italics added)

In exploring a romanticized East-West Binary Fenollosa is exhibiting what pretty much exactly what Edward Said critiqued as the heart of colonial Orientalism. However, here we can also see the importance which Fenollosa places on the "Spiritual Wisdom" of the East and its vital role in world history. The ultimate goal of the Spirit is a merger of East and West, which will blend western science and the eastern spirit. What the East has to offer the West is its religious tradition, which is exemplified by its "spiritual knighthood" and "esoteric discipline".

Fenollosa saw both of these epitomized in Tendai Buddhism. As he remarked: "This great esoteric sect, which ascribes magical power and direct contact with the spirit to the human soul, was called, from its central sects, the Tendai sect. The mastery of self, the spiritual knighthood which it preached [were popular]." ([Fenollosa 1912](), p. 121). Just as Okakura seemingly saw himself as a ronin-scholar, Fenollosa seems to have seen himself as a Tendai spiritual knight and indeed, embraced Tendai practice in his personal life.

Not surprisingly, Fenollosa finds "spiritual knighthood" personified in his Tendai teacher. This is fully explicit, as Fenollosa writes: "His Reverence the Archbishop Keitoku, of the Tendai sect at Miidera Temple on Lake Biwa, I still look up to as my most inspired and devoutly liberal teacher in matters religious . . . He was a lofty exemplar of the spiritual knighthood." ([Fenollosa 1893](), p. 211). Archbishop Keitoku occurs as an exemplar throughout Fenollosa's thoughts and serves as a mouthpiece for his ideas about the Buddhist path. He claims the following poem was Keitoku's central teaching:

> Must first of all conquer himself;
>
> The true knight
>
> With his own heart fight . . .
>
> On no external god relying,
>
> Self-armed, heaven and hell alike defying,
>
> Lonely,
>
> With bare will only,
>
> Biting his bitter blood-stained sod; —
>
> This is for the *world*, as for Japan,
>
> This is to be a man!
>
> This is to be a god! ([Fenollosa 1893](), pp. 28, 212)

Again, Tendai Buddhism is a spiritual knighthood, here understood as an internal conquest (although colonialist imaginary is probably not too far in the background). Further,

Fenollosa suggested elsewhere that the Tendai meditative practices are the method for engaging in an internal struggle for self-knowledge. He writes:

> The young soul had to win the spurs of its knighthood alone, in struggle, in effort to feel and see, in invocation to the gods to tear his heart open—alone before the altar in his cell, or his own chamber shrine . . . You gaze into the white, round mirror on which is painted in Sanskrit the golden breathing "ah-h!" and you watch while its surface deliquesces, expands to an infinite crystal sphere . . . (Fenollosa 1912, p. 156)

Fenollosa is describing a practice common to both Shingon and Tendai esoteric traditions of focusing upon the Sanskrit syllable *A*. Here *A* is used as the root mantra for a series of visualizations which serve as a preparatory step for the larger esoteric sequence (see Payne 1998). The struggle for self-knowledge seems to be located in the esoteric environment of mantra and concentrated visualizations.

The surprise might be that Fenollosa's idea of meditation as an internal battle is not totally without precedents. Some of these themes can be read in the *Shōshikan* 小止観 attributed to the Tendai patriarch Zhiyi 智顗, which is also the only Buddhist text that Okakura translated into English. The *Shōshikan* is one of the classic meditation texts of the Tendai tradition, often used as introductory manual or sometimes for lay practitioners (See Bielefeldt 1988, p. 63). The original is fairly unique as a meditative text because of the amount of space which is devoted to the discussion of the dangers of meditation and the powers of *mara* (Japanese *ma* 魔). However, because of the selectiveness of Okakura's translation, themes in the portion of the text are exaggerated and Okakura's version gives the sense of an opposition, not between two kinds of practice (good and bad) but between two forces (good versus evil) which battle within the practitioner. For example, the original has two chapters entitled 善根發 and 覺知魔事 which Okakura translates as "Growth of Powers for Good" and "Knowing the Evil Spirits" (more literal might be "Establishing the Roots of Virtue", and "Recognizing the Activities of Mara").

Take together what this suggests is that Okakura and Fenollosa were not only key figures in the invention of Japanese fine art, but they were also equally complicit in the formation of a particular kind of Japanese Buddhism which was also radically reconfigured via in relation to Euro-American religious (and philosophical) concerns. The final point I want to underscore is that even as these two men had a modest impact on these global categories, they themselves were transformed in turn. Indeed, after his death in 1908 Fenollosa's ashes were buried by Lake Biwa in the grounds of the Miidera temple where he had taken his Bodhisattva vow. In some metaphorical sense, he traded himself for the Bodhisattva he had removed from Yumedono so many years previously—one body buried and one body removed from its crypt.

## 4. Conclusions: The Uses and Abuses of Analytical Categories

As Horii Mitsutoshi has helpfully observed, much of the current debate is about "'religion' as an analytical concept in the Japanese context" (Horii 2020). To this we might productively ask: if "religion" is indeed "an analytical concept", what kind of work do analytical concepts perform? What are they good for? What makes something a better or worse analytical concept?

The answers to these questions are not obvious, and they are of fundamental importance if we want to have something to take away from this endeavor beyond the instrumental utility of grant funding. If we were merely doing thick description or philological reconstruction, we could probably avoid having to produce general theoretical edifices, but the very nature of broader research means that these should not be ignored. Hence, the earlier we figure out the philosophical stakes of ordinary scholarly praxis, the better it will be for the overall trajectory of our collective endeavor.

The contemporary Dutch scholar Rens Bod has argued that the human sciences can be characterized in terms of "the quest for patterns in humanistic material on the basis of methodical principles" (Bod 2014, p. 7). I think this is roughly correct, as far as it

goes. Much of what we think of as "analytical concepts" or theories are assertions about patterns, which are necessarily rooted in relations of similarity and difference. However, this opens a particular problem that has been the bane of many theoretical programs across the disciplines, namely that pattern recognition alone does not tell us how and where patterns apply.

I go into a lot more detail about this elsewhere, but in brief, stronger generalizations (which are really abductive inferences) are those that present a hypothesis about the relationship between the observations and the cause of those observations, which in turn necessitates a theory about the degree to which the sample is representative. We can become a lot better at understanding generalizations like "crows will be black", if we have a theory that provides a causal explanation behind the coloring of crows.

This has concrete implications for how we formulate analytical concepts. In order to rise to the level of explanation, we need to specify common properties *and* the causal, anchoring processes that have produced those properties. That is what I aimed to provide above with my gesture toward anchoring processes.

At the first and broadest level, the theory sketched out above gives us an important clue about the utility and limitations of "religion" as an analytical concept. In summary, part of the problem with unconsidered usages of the category "religion" is that they presume that there is a natural kind classification which amounts to "religion" and which provides "religions" with their common features. However, as I have been suggesting, "religion" is not a de facto natural kind. Neither is it an ergonic kind. There is not one single need that motivates all religion (or all art for that matter). More than a century of scholars searching for such have failed. Rather, to use the vocabulary above, "religion" is a dynamic-nominalist social kind. This amounts to the claim that religions mainly only come to share properties in common because they have been classified as such.

Recast, "religion" in its contemporary formulation is largely a diplomatic and legal category, and not just an academic or ethnographic one; it is these processes that largely anchor its attributes or police its boundaries. This means that we are able to make more robust generalizations about "religion" only in the contemporary period once those classifications and legal mechanisms have been brought into place.

To go into one more level of detail, the virtual neologism *shūkyō* was coined in Japan and popularized throughout East Asia. It partially replaced a previous set of equally dynamically nominalist second order categories (e.g., *buppō* 仏法), which had their own classificatory logics and institutionalized boundary policing. As "religion" came to the fore, it brought with it a set of conceptual distinctions. Moreover, Christianity served as a prototype for the category and brought with it a whole baggage of ancillary terms and their implicit distinctions: faith versus knowledge, prayer versus magic, bifurcation between religion and science, religion versus superstition, and, this is important, between, religious" and the secular. All that is to say, being classified a "religion" changes the entities thus classified. It introduces distinctions not previously available, and it produces connections that were not previously on offer.

In summary, the problem with the category of religion is more than just a translation issue. The introduction of the concept of "religion" results in visible and far-reaching changes—intellectual, legal, and cultural. In short, the category has spread to the point where discourse about "religion" can now be found in every region of the globe, thanks to centuries of self-identifying secularists or religionists, as well as colonial administrators, diplomats, missionaries, subalterns, scholars, newspaper editors, and anyone else who joined the conversation in any given language. This has resulted in huge institutional transformations and legislative upheavals, as well as a new discourse about "religion". Few people can avoid having an opinion about religion; such an opinion, which was practically nonexistent before the year 1700, has now become nearly ubiquitous. Concurrently, the category has become so diverse that any pretense of universal agreement on religious bounds can no longer be maintained. The only matter anchoring whatever limited generalizations

about contemporary "religion" that are currently possible is a set of legal and discursive structures that in the late nineteenth century came to encircle much of the globe.

Although differing in the particulars, the invention of "fine art" was the result of similar processes. This parallel is no accident. This is a clue that the histories of "religion" and "fine" are not two independent narratives, but rather the tale of entangled categories born late and in the moment of their mutual contrastive self-definition. Over the series of the four volumes that established his reputation, the economic historian Immanuel Wallerstein argued that the current global system of commerce and ideas emerged in the long 16th century before fully entangling the globe by about 1900. It might seem to be no accident that this iteration of the world-system overlaps with what Reinhart Koselleck refers to as the "saddle period" (*Sattelzeit*), during which most contemporary social and political concepts were formulated. Taken together, Wallterstein and Koselleck's research would suggest that by the end of 19th century, the globe was more or less entangled into a *single* system of capitalistic commerce and to a lesser extent modes of discourse. "Fine art" and "religion" (as second order categories) were globalized in this historical period and as part of the production of this economic and cultural system. Although Japan entered this world-system in a staggard manner, it is no surprise that it had to indigenize these master categories in the same period. However, this is not all we can say about them.

To go back to the beginning, thinking about the disciplinary master categories as processes of kinds changes our orientation them. Again, I have written books (and could write more) on the topic of "religion". First, because "religion" is often essentialized, we must begin by de-reifying it. To consider religion as a social activity is to first grant the critique. Religion is a culturally conditioned concept with a disturbing, colonial history that generally brings with it a set of unjustified assumptions. There is no core, no essential and sufficient prerequisites for membership in "religion". Instead of the binaries established by essentialism's naïve presumption, we may now speak in degrees—for instance, objects labeled "religion" might be both colonial and anti-colonial in various ways in different locations.

Furthermore, we must recognize that "religion" is not a cross-cultural universal, but rather a collection of particulars. Religion, as a result, acts differently as a social kind in different cultures, times, and discourses. Its distinction from other fields, such as the secular, is erratic and unstable. Religion has always been historically dependent. The same may be said of "religions".

That said, while conditioned, the construction of "religion" is not arbitrary. It has been reified, yet the process of reification may be examined. Religion is relational, but it is far from nonexistent. It is a movable signifier, yet it has a specific grammar. It is a byproduct of a historical moment, yet it cannot be reduced to its source. It exists as discourse, yet it has components that defy incorporation. It is the result of hegemonic power, but it is also a place of opposition.

There is a lot more to be said here. However, most items labeled as "religions" only share abilities with other category members as a result of dynamic-nominalist anchoring processes, as an examination of this topic indicates (most of their internal features are shared owing to mimetic processes). Various entities are designated as "religions" by being named or categorized as such by internal diplomacy, domestic legislation, scholars, and so on. This is not a teleological or transhistorical process, but one that arose from a certain logic at a specific time in Western Christendom, and its globalization was inherently selective and, to some degree, arbitrary. Furthermore, rather than being affected by unidirectional force or hegemony, it was negotiated. Moreover, as I have established, there must be multiple incentives or enforcement mechanisms in place to encourage the categorization to take hold.

Both categories were catalyzed by the social differentiations of industrial civilization— think of contemporary American cities with their split between places of commerce, homes, factories, schools, museums, places of worship (further Shinto versus Buddhist versus Christian), and so on. To be clear, this was not the differentiation of a previously undif-

ferentiated space. Pre-Meiji Japan had its own sets of distinctions, 寺神社, shrine-temple complexes, temple markets, temple schools, craft guild spaces, and the like, but the organizational logic was different. The encounter with the world-system of the late nineteenth century did not merely replace one organization logic with the other, but was a complex, multi-generational process of negotiation or contestation, producing new hybrid spaces, returns, transformations, and innovations, even such that they reflected back on other global systems, changing them in subtle but profound ways.

The Boddhisattva of the Hall of Dreams serves as a case in point. As several art historians have observed, the statue has mid-seventh-century characteristics and materials (see Weistein). As a process social kind, the statue seems to have been produced both mimetically (in imitation of the Eastern Wei style, which reached Japan via Korea in roughly the same period) and dynamic-nominalist as well, insofar as it was intended to be a member of the category "Guze Kannon" (etcetera). These are different anchoring processes than those that produced the category "religion" and its boundaries. However, and this is an important point, the social properties of the Yumedono Kannon have changed and continue to change over time. For instance, in the Kamakura period, it gained a reputation as a *hibutsu* and as such it gained the attendant social properties of that category. By the nineteenth century, it was classified as "fine art" and, as Fenollosa's case illustrates, therefore came to be experienced in different terms. All that is to say, the meaning of the same material object changed over time.

All that is to say, the question is not "*is this statue religious*?" but "*when is this statue religious*?" and social kind theory gives us the capacity to answer that. A statue displayed in an art gallery is going to be read as an art object with some of the attendant attributes of art objects (e.g., people will attribute to it a specific creator, attempt to read it aesthetically or formally, or in its broader art historical context, etcetera). If the same statue is placed on an altar in a temple, it is then going to take on a different set of attributes (e.g., people will tend to see it in terms of its symbolism, of say Buddhist doctrines, and so on). In summary, a specific artifact's membership in a broader set of kinds is situationally and contextually dependent. Moreover, it is never complete as the other meanings and contexts continue to be available.

The last account of the Bodhisattva of the Dream Hall, and its possible meanings, has yet to be written.

**Funding:** This research received no external funding.

**Institutional Review Board Statement:** Not applicable.

**Informed Consent Statement:** Not applicable.

**Data Availability Statement:** Not applicable.

**Acknowledgments:** I would like to thank Horii Mitsutoshi for encouraging me to contribute this article to the special issue. Section 3 above includes material I presented in a seminar in graduate school taught by Carl Bielefeldt in 2004. Thanks are due both to his feedback and permissiveness in letting me pass in such a crude work, which I hope I have improved here ... Thanks are also due to suggestions by Pamela Winfield, Tsuchikane Yasuko, and Chinghsin Wu and those that attended our Zoom roundtable on "'Religion' and the Modern Artist in 20th Century Japan" at AAS 2021.

**Conflicts of Interest:** The author declares no conflict of interest.

## Notes

1    See https://www.tnm.jp/modules/r_free_page/index.php (accessed on 17 January 2022).

2    To be clear, there was also an older Chinese category of "literati painting" or *wenrenhua* (文人畫), which, starting in the 16th century, was retroactively projected backward in some areas to demarcate a kind of elite rather than popular art. However, "literati painting" was very different from a notion of "fine art" insofar as it was focused on the practice of painting as a form of self-cultivation for the scholar-poet rather than as a product. What made it elite art was not what it looked like, but who produced it. See (Clunas 2017).

3 To prevent misunderstanding, "process social kinds" are not *social* in contrast to cultural, political, artifactual, economic, or symbolic kinds. My notion of process social kinds is intended to be a higher-order category that would include all those kinds as well. The term could be thought of as shorthand for "socially constructed kinds".

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
