# Peer review of "Excavating the Hall of Dreams: The Inventions of “Fine Art” and “Religion” in Japan"

_religions, doi:10.3390/rel13040313_

Round 1
Reviewer 1 Report
The thesis of this paper is unclear. In the abstract, lines 10-12 constitute the thesis, but the meaning of the main sentence there is unclear: “It will argue that attending to these parallel co-constructions of “religion” and “art” will help 10 shift the debate in critical religion off of an obsession with translation and onto more fundamental 11 issues about continuity and rupture of "cultural categories.'" I do not know exactly what this sentence means. I would recommend a shorter and more concise sentence here.
In the main text, the thesis appears in lines 48-56, but it is again unclear. The reader understands that there have been debates in relationship to the hidden idol, possibly as pertains to whether the item is of a religious or artistic nature. The author needs to clearly state the thesis.
Lines 233-236 present more of a simple thesis for the paper, though the wording should be revised and should appear earlier in the paper.
The paper has some very interesting material. However, there appear to be different papers happening at the same time. One topic pertains to the excavation of the Hall of Dreams, which is also referenced in the title and introduced at the beginning in the discussion of the discovery of the Bodhisattva of the Hall of Dreams. However, the paper then moves onto other topics, including: 1. the problem with the category of "fine art" or "art", 2. the reception of Japanese art in Europe, 3. the problem with the category of "religion" and 4. the problem with categories in general when conducting research in the humanities.
There could be a way of bringing these topics together more clearly but a main topic should emerge. One of the stronger topics has to do with the problem with the category of "religion." Here, the author would need to reference the seminal work of Jonathan Z. Smith, whose essay, "Religion, Religions, Religious" (1982/3) was the seminal work that brought up the problem with the category of religion within the field of religious studies. In the context of East Asian religions, Vincent Goossaert's 2005 essay "The Concept of Religion in China and the West" is also important.
Similarly, the paper also discusses social ontology but makes no reference to the main theorists who have worked on it in philosophy (John Searle) and religious studies (Kevin Schilbrack).
I would suggest bringing forth more prominently the topic related to the excavation of the Hall of Dreams and making the discussion about the problems of categories shorter.
Author Response
I appreciate the reviewer's suggestion that I should improve the clarity of the thesis. I have done so.
I talk about social ontology in a lot more detail (and engage more fully with the secondary literature including Searle and Schilbrack) elsewhere and I think it would make the essay seem even more disjunctive to go on an even longer discussion in part 3. So I've radically cut part 3. And cut almost all references to social ontology.
J.Z. Smith did not initiate the critical turn that was Wilfred Cantwell Smith decades earlier. But I've added a footnote to Goossaert.
I've generally trimmed part 3 by more than 1/2 to bring the whole thing greater coherence.
Reviewer 2 Report
I do not have comments related to the content of the paper. It is well-documented, it transpires that the author/s is/are knowledgeable and expert in their field. The thesis is evident; thus, the analysis unfolds and leads the reader/s to access the knowledge evidenced in the article.
However, I have comments that refer to the format, which diminishes the quality of such an excellent paper. I have five comments in this regard:
- On page 1, a direct quotation goes over four paragraphs (lines 25-46) set as regular paragraphs. The quote exceeds the forty-word rule yet is not indented as a block. It is easily confused as the author/s' work and not as a quotation from other author/s. Editors should correct that. For example, the same happens with quotes in lines 190-194 and 402-407. Concurrently, the author/s do not engage in length that extensive quotation, which makes me wonder about the purpose of transcribing it in the article.
- On page 2, footnote 1 is not superscript. At the same time, using in-text citation requires all references to appear in a single list at the end of the article, reserving the footnotes only for clarification and not for references. Therefore, that information needs to appear in-text in the short form and the complete reference in the list of references. Finally, when clicking on the URL (which should appear in the reference list), it does not exist on the web.
- I would also recommend to the author/s to split sentences into shorter ones. Some of the sentences exceed the usual "twenty-five words" rule, making it difficult for the reader/s to follow the analysis.
- Although the author/s divided the article into sections, those sections are pretty extensive. I would recommend including subheadings in each section to help the reader/s navigate the paper.
- Finally, it is customary to include the abbreviation of US States' names (i.e., CA =California) after the city. Some cities have abbreviations in the reference list, and others do not.
Author Response
Great. Suggestions. I've tried to trim some of the long sentences and I've added in subheads to clarify. I've also radically tightened section three. In the final version (once I have a better sense of the journal's standards) I'll fix the bibliography.
Reviewer 3 Report
Your analysis of category formation is very apt and will be of use to others in the field. Excellent contribution.
This essay is very well written, clear, and concise. The argument made is novel, particularly in drawing into conversation the development of. “religion” as a new category in 19th century Japan with the development of the category of “fine art.” Category formation is a very critical aspect of historically locating the study of religion, and is key to avoiding imposing distorting preconceptions. This essay provides both concrete information about category formation and useful theoretical reflections. This essay would work very well for a graduate seminar in either Japanese religions or religious studies.
Author Response
Great. Thanks for this reader's kind and perceptive comments! Much appreciated.
Round 2
Reviewer 1 Report
The thesis of this paper is better but still needs to be developed. The abstract in this revision is better than it was in the first draft. At the end of the Introduction, the author states, “This article will set off from these debates because it is my contention that paying attention to the temple and the museum as major venues for the local contestation of globalizing categories will help us make progress in critical religion by helping us pinpoint problems with the analysis of cultural categories.” This sentence is too long and the meaning is unclear. Smaller sentences and clear, meaningful words would make it easier for the reader to understand the direction of the paper. The conclusion is much more clear (paragraph on page 16 beginning “All that is to say, the question is not “is this statue religious?” but “when is this statue religious?”) and could be utilized to bring the same ideas into the beginning (and thesis) of the paper.
The author states that the paper will “advance the debate by building off of a process social ontology.” On page 12, the paper then makes reference to “process social ontology” with the statement, “A detailed explanation is impossible due to space constraints. I would direct readers elsewhere for further explanation of what I have called, a “process social ontology” approach to “social kinds” (see Storm 2021).” There is no further explanation of “process social ontology.” A reference to “social kinds” seems more appropriate and could be used in the paper instead of “process social ontology.”
Jonathan Z. Smith’s ideas on the problem of the category of religion should be referenced in this paper. The author references Talal Asad, Wilfred Cantwell Smith and Bret Nongbri on this account but does not mention JZ Smith. Nongbri’s work is highly influenced by Smith’s. One of the reasons that JZ Smith is considered integral to the modern discipline of religious studies is connected to his discussion of the category of religion. In my earlier review of this paper, I mentioned the importance of referencing JZ Smith in the context of this paper’s topic. The author stated that Wilfred Cantwell Smith predates Smith, but Smith’s own contributions are still considered seminal. At the very least, Smith should be mentioned.
Other cultures, including those of China and East Asia, had their own traditions of dividing what might be considered “high art” from the work of artisans. In China, literati painting was historically revered as the higher form of art from other painting and artwork. These divisions, which pre-dated Europe’s involvement in Asia and were indigenous, should be mentioned.
The strength of the paper lies in the focus on how objects change meaning depending on the social context in which they are used.
Author Response
I have now responded to these excellent suggestions:
1) I've revised the offending sentence in the introduction to make it clearer. It now also mentions that the paper will show how objects change meaning depending on the social context in which they are used.
2) I've cut all but one reference to process social ontology and focused on social kind instead.
3) I've mentioned J.Z. Smith and added bibliographic reference to his work.
4) I've added a footnote on literati painting (文人画) as an indigenous Chinese category.